# Clinical Characteristics of Patients with Intraocular Lens Calcification after Pars Plana Vitrectomy

**DOI:** 10.3390/diagnostics13111943

**Published:** 2023-06-01

**Authors:** Silvia Bopp, Hüseyin Baran Özdemir, Zeynep Aktaş, Ramin Khoramnia, Timur M. Yildirim, Sonja Schickhardt, Gerd U. Auffarth, Şengül Özdek

**Affiliations:** 1Capio Augenklinik Universitätsallee, 28213 Bremen, Germany; 2Department of Ophthalmology, Gazi University School of Medicine, Ankara 06570, Turkey; 3Department of Ophthalmology, Atilim University School of Medicine, Ankara 06830, Turkey; 4The David J Apple International Laboratory for Ocular Pathology, University of Heidelberg, 69117 Heidelberg, Germany

**Keywords:** IOL calcification, IOL opacification, pars plana vitrectomy, endotamponade, cataract surgery, IOL exchange

## Abstract

Aim: To determine the clinical risk factors that may increase the occurrence of intraocular lens (IOL) calcification in patients who had undergone pars plana vitrectomy (PPV). Methods: The medical records of 14 patients who underwent IOL explantation due to clinically significant IOL opacification after PPV were reviewed. The date of primary cataract surgery, technique and implanted IOL characteristics; the time, cause and technique of PPV; tamponade used; additional surgeries; the time of IOL calcification and explantation; and IOL explantation technique were investigated. Results: PPV had been performed as a combined procedure with cataract surgery in eight eyes and solely in six pseudophakic eyes. The IOL material was hydrophilic in six eyes, hydrophilic with a hydrophobic surface in seven eyes and undetermined in one eye. The endotamponades used during primary PPV were C2F6 in eight eyes, C3F8 in one eye, air in two eyes and silicone oil in three eyes. Two of three eyes underwent subsequent silicone oil removal and gas tamponade exchange. Gas in the anterior chamber was detected in six eyes after PPV or silicone oil removal. The mean interval between PPV and IOL opacification was 20.5 ± 18.6 months. The mean BCVA in logMAR was 0.43 ± 0.42 after PPV, which significantly decreased to 0.67 ± 0.68 before IOL explantation for IOL opacification (*p* = 0.007) and increased to 0.48 ± 0.59 after the IOL exchange (*p* = 0.015). Conclusions: PPV with endotamponades in pseudophakic eyes, particularly gas, seems to increase the risk for secondary IOL calcification, especially in hydrophilic IOLs. IOL exchange seems to solve this problem when clinically significant vision loss occurs.

## 1. Introduction

Opacification due to calcification of intraocular lenses (IOL) is a rare but severe complication and may need IOL exchange [1]. Calcification of hydrophilic acrylic IOLs has been a well-known phenomenon since 1999, but the mechanism is not fully understood yet. Calcification is classified as primary, secondary and pseudo-calcification [2]. Whereas primary opacifications are inherent to the IOL itself, mainly to the IOL material, manufacturing or packaging problems [3,4], secondary opacifications are thought to occur as a result of environmental factors, such as ocular inflammation and foreign materials in the anterior chamber like air, gas or silicone oil and were suggested to increase the risk of IOL calcifications [5,6].

Opacification has been more frequently reported in hydrophilic acrylic IOLs. Hydrophilic acrylic IOLs with hydrophobic surface were designed to decrease primary calcification, but the occurrence of calcification in these IOLs has also been reported recently [4,7]. Increasing numbers of cataract surgeries and secondary interventions such as posterior lamellar keratoplasty (DMEK and DSAEK) necessitating intracameral injection of air or gas also resulted in growing numbers of IOL opacification cases. Another risk factor is pars plana vitrectomy (PPV) with gas tamponade leaking into the anterior chamber. IOL calcification was previously reported to occur following PPV with sulphur hexafluoride (SF6), perfluoropropane (C3F8) and air [8,9,10].

IOL opacification may cause a decrease in visual acuity and contrast sensitivity [11,12]. Significant opacification of the IOL optic causes visual deterioration, and the only treatment is IOL explantation. As the capsular bag cannot be opened and the diaphragm cannot be preserved in all cases, secondary IOLs must be placed in the sulcus (with or without suturing) or fixed to the iris (iris claw type anterior to or behind the iris).

Here, we report clinical findings of calcified IOLs that occurred following PPV and were explanted due to progressive visual impairment. The aim of the study was to investigate the factors that may increase the risk of calcification and the time interval between PPV and IOL opacification.

## 2. Materials and Methods

This retrospective study included 14 patients who underwent IOL explantation due to clinically significant IOL opacifications after PPV in two tertiary referral centres (Bremen Eye Clinic Universitätsallee, Bremen, Germany and Department of Ophthalmology, Gazi University, Ankara, Turkey) between June 2015 and July 2017. Extracted IOLs were sent to and analysed at the David J Apple International Laboratory for Ocular Pathology, Department of Ophthalmology, University of Heidelberg. The study protocol was approved by the local Ethics Committee of Gazi University. The study was conducted in compliance with the tenets of the Declaration of Helsinki. Informed consent was taken from all patients before surgery.

The medical charts of all patients were reviewed in detail. The following data were collected from medical records: age, sex, systemic diseases, coexisting ophthalmic diseases, the date of IOL implantation, implanted IOL design/material/brand/dioptre, the date of vitrectomy, vitrectomy indication, vitrectomy manoeuvres and tamponades, presence of the postoperative intracameral gas, additional surgeries after vitrectomy, the date when IOL opacification was recorded first, location of IOL opacification, the time interval between vitrectomy and IOL opacification, the time interval between IOL opacification and IOL explantation, the new IOL design/material/brand/dioptre implanted, best corrected visual acuity (BCVA) after vitrectomy, prior to IOL explantation and after IOL exchange, posterior capsule data such as previously performed YAG-laser capsulotomy or intraoperative capsulotomy.

A review of the reports of the previous surgeries has revealed that either conventional 20-gauge or transconjunctival 23-gauge PPV had been performed as a single procedure in pseudophakic eyes or as combined surgery with phacoemulsification and in-the-bag IOL implantation in primary phakic eyes. Air, C2F6, C3F8 or silicone oil had been used as endotamponade agents.

IOL explantation was performed in patients who had a loss of vision due to IOL calcification. The first step of the surgery was trying to free the IOL from the capsular apparatus. If capsular adhesions were too firm and/or the capsular bag was already weak, the IOL and capsule were extracted together through a corneoscleral incision. A new three-piece IOL was implanted into the sulcus in the eyes with sufficient capsular support. If an adequate capsular diaphragm was not present, the three-piece IOL was sutured to the sclera or an iris-fixated IOL was used.

All intraocular lenses were sent for analysis by the surgeon to the David Apple International Laboratory for Ocular Pathology, University of Heidelberg, Germany. The methodology of analysis is described in detail by Yildirim et al. [13]. In short, after microscopic examination, the IOLs were stained with Alizarin red for detection of superficial calcium deposits. Of note, 5 µm vertical cross sections from the optical centre of the lens were stained using the von Kossa method to find subsurface calcium deposits [13].

Statistical analyses were performed with SPSS 22.0 (IBM, Armonk, NY, USA). The Student *t*-test was used to compare Snellen visual acuity results.

## 3. Results

The demographic data of 14 eyes of 14 patients with IOL opacification are presented in Table 1. The mean age of patients was 54.10 ± 15.50 and the female/male ratio was 6/8. Thirteen out of 14 IOLs were made of hydrophilic acrylic material (7/13 with hydrophobic coating), and one IOL’s specifications were unknown but showed the same opacification pattern. Six out of 14 IOLs had plate haptics, seven of 14 IOLs had C-loop haptics, and one IOL had prolene haptics. The mean IOL dioptres were 18.23 ± 3.34 D (range: 13.00–26.50 D).

Eight eyes had undergone simultaneous phacoemulsification, IOL implantation and PPV surgery (Table 2). The remaining six eyes were pseudophakic, and PPV was performed after a mean of 43.4 months after cataract surgery (range: 2–108 months). Indication for PPV was pseudophakic rhegmatogenous retinal detachment (RRD) in six eyes (42.85%), phakic RRD in six eyes (42.85%), tractional retinal detachment in a uveitis patient (7.15%) and macular surgery for an epiretinal membrane in one patient (7.15%). C2F6 gas tamponade was used in eight eyes, C3F8 was used in one eye, air was used in two eyes, and silicone oil was used in three eyes. One patient who was given an air endotamponade underwent repeat vitrectomy with silicone oil tamponade.

Patients with silicone oil underwent subsequent silicone oil removal and gas tamponade with C3F8, SF6 or air. Postoperatively after primary vitrectomy, intracameral gas was detected in six eyes (42.85%). Two patients required YAG-laser iridotomy for a pupillary block due to gas in the anterior chamber, and two patients underwent anterior chamber revision with gas removal. Five patients had received an intraoperative posterior capsulotomy during primary surgery with air/gas, and another one had already a YAG-capsulotomy before RRD surgery. An additional three eyes received Nd:YAG-laser capsulotomy in the later postoperative course. Including silicone oil removal, additional vitreoretinal procedures were performed in six patients. One of six patients underwent four consecutive vitreoretinal surgeries. Repeat vitrectomy was performed in another two patients, and one of these patients underwent silicone oil removal later. The remaining three patients underwent further silicone oil–air/gas exchange.

The clinical data before and after the detection of IOL calcification are presented in Table 3. The mean interval between the first PPV and documented IOL opacification was 17.8 ± 15.4 months (2–54 months). The mean interval between first air/gas contact with IOL and documented IOL opacification was 16.90 ± 15.70 months (2–54 months). The mean LogMAR BCVA after PPV was 0.43 ± 0.42 (0.00–1.30), and the mean BCVA before IOL explantation due to IOL opacification was 0.67 ± 0.68 (0.1–2.70). The decrease in BCVA was statistically significant (*p* = 0.007). The mean BCVA increased again to 0.48 ± 0.59 LogMAR (0.00–2.30) following the IOL exchange. The increase in BCVA after IOL exchange was statistically significant (*p* = 0.015). There was no statistically significant difference between the initial BCVA before IOL calcification and the BCVA after the IOL exchange, indicating that the mean vision returned to the original level following the IOL exchange (*p* = 0.905).

Central opacification of IOLs was demonstrated at the slit-lamp examination (Figure 1 and Figure 2). Nine of 14 IOLs (64.28%) had opacification at the anterior surface, and two of 14 IOLs (14.28%) had opacification at the posterior surface. The David J Apple International Laboratory for Ocular Pathology examined and identified calcium phosphate deposits in all IOLs located within the pupillary area, always sparing the haptic and sometimes the peripheral optic (Figure 3).

## 4. Discussion

The present study focused on the clinical data of the patients who underwent IOL explantation for opacifications that occurred after PPV. We assume that there are many unrecorded or misdiagnosed cases, the latter receiving YAG-laser capsulotomy for the misdiagnosis of after-cataract. The opacifications are caused by superficial calcium deposits at the surface of the IOL. They are located in the pupillary area, usually at the anterior surface, less common at the posterior surface. Calcifications can occur in a very short time period (within two months) or may be seen as a late complication (after 54 months) [5,14,15]. Visual deterioration was recorded on average 20.5 months after PPV in our study. Analysis of clinical data indicates that intravitreal gas or air tamponade increases the risk of IOL calcification in eyes with both hydrophilic IOLs and hydrophilic IOLs with a hydrophobic surface. The presence of gas/air in the anterior chamber and a posterior capsulotomy that enables direct contact of intravitreal gas/air to the IOL surface seems to be a predisposing factor for IOL calcifications. Some of our patients had primary vitrectomy with silicone oil and developed IOL calcifications thereafter. When reviewing the medical history, all eyes had at least once gas/air in the eye during the postoperative course. Thus, the beginning of the structural changes in the IOL was most likely attributable to the air/gas and not to the primary silicone oil tamponade.

There are very few cases published reporting IOL calcification after PPV with silicone oil tamponade [16,17,18]. In the majority of them, IOL opacification occurred after silicone oil removal. In view of known risk factors, it cannot be judged with certainty whether silicone oil itself or accompanying factors (see below) provoked calcification. We have observed a few diabetic patients with a silicone oil tamponade and some central calcifications at the lens surface who did not accept lens exchange which was followed for a couple of years without significant progression (Figure 4).

Calcification of IOLs is a known phenomenon first reported by Apple et al. in 2000 [19]. The reports and laboratory analysis of the explanted IOLs showed that the opacifications consist of calcium phosphate/hydroxyapatite deposits at the IOL surface [13]. The exact pathophysiology leading to those accumulations on the IOL surface remains a matter of discussion. Supersaturation of humor aqueous with calcium may lead to precipitations at the IOL and initiate deposition of microcrystals that may enlarge thereafter. It occurs with hydrophilic acrylic IOLs with hydroxyl groups at the surface and can bind calcium compounds, but hydrophilic IOLs with hydrophobic coating may also show calcification. The causes and the alterations of the intraocular milieu triggering the deposition of calcium phosphate are not known. It is hypothesized that foreign material, such as a gas tamponade, intraocular inflammation or specific topical medication may initiate this process. Most likely the mechanism is multifactorial [3,15,20].

After the 2000s, manufacturing-related calcifications emerged in some specific types of IOLs, which were classified as being primary calcification [2,21,22,23,24]. Material and manufacturing processes, as well as packaging and storage problems, may increase the risk of calcium and phosphate depositions into the IOL [1]. Manufacturers withdrew those lenses from the market and changed manufacturing or packaging methods. Furthermore, a coating of the hydrophilic material with hydrophobic material was developed to overcome this problem. However, the calcification of these coated IOLs has also been observed [4,7]. Our patients had various types of IOLs with different designs and materials (Table 4). Thirteen of 14 IOLs in our series were hydrophilic, and interestingly, seven of them had a hydrophobic coating. Twelve of 14 IOLs were single-piece IOLs (plate haptic and C-loop design), and one of the IOLs had a three-piece design. Therefore, the risk of calcification is increased in the hydrophilic IOL material but not affected by the IOL design.

As secondary calcifications of hydrophilic IOLs are extremely rare after uneventful cataract surgery, other factors than the IOL material alone must be causative. One particular finding is the presence of intraocular air/gas with temporary contact with the IOL, which was a uniform finding in our patients. In analogy to that, there are several reports of IOL calcifications that occurred after posterior lamellar keratoplasty [5,25,26,27]. The use of air or gas for the attachment of the graft is thought to be associated with IOL calcification in the presence of a hydrophilic acrylic IOL. Repeat intracameral gas or air injections (rebubbling) increases the risk of IOL calcifications [28,29]. The same is valid for other clinical situations, such as intracameral gas or air injection for Descemet membrane detachment, for wound sealing after cataract surgery and gas dislocation from the vitreous cavity into the anterior chamber [30,31]. In total, at least eight out of 14 eyes had contact of gas to the anterior or posterior lens surface via a capsulotomy or due to gas dislocation. Thus, the presence of gas and probably direct contact with the IOL can be considered as an important trigger for secondary IOL calcifications. Recently, Marcovic et al. analysed 11 cases with similar clinical characteristics [32]. All had undergone vitrectomy and gas tamponade for either macular diseases or RRDs. The authors speculated that intraocular gas might dehydrate the hydrophilic IOL, irrespective of direct contact of gas with the IOL or capsule and then leading to changes in the IOL surface that make it prone to calcium phosphate sedimentations. As most reports did not analyse details of the lens capsule or a possible gas prolapse, the role of a posterior capsulotomy could not be assessed with certainty at present [9,10,14,33].

Surgical inflammation caused by vitrectomy, retinopexy and tamponade agents leads to a breakdown of the blood–aqueous barrier to some extent in all eyes. Most cases of IOL calcification after PPV were reported to have previous combined phacoemulsification and vitrectomy [9,16,17,18,34,35]. Even if clinically not relevant, combined phacoemulsification and vitrectomy is accompanied by more postoperative inflammation compared to vitrectomy alone [35]. Similarly, eight (57%) of the cases in our series had undergone combined surgery. Although secondary IOL calcifications are extremely rare after uneventful cataract surgery, cases with enhanced-inflammation-received intracameral rt-PA (recombinant tissue plasminogen activator) to treat fibrin formation were shown to develop calcium depositions [36], which stresses the fact that enhanced inflammation is an essential trigger for calcium deposition.

Inflammation may be even more pronounced in diabetes-related ocular pathologies [37]. Park et al. reported five cases of IOL calcification in eyes with proliferative diabetic retinopathy-related vitreous haemorrhage that underwent PPV [34]. However, there were no patients with diabetic ocular changes in our cohort and none in the series reported by Marcovic [32].

Additional vitreoretinal procedures such as repeat vitrectomy or silicone oil removal, which were performed in 43% (six eyes) of our patients, may induce another episode of intraocular inflammatory reaction. The risk of IOL calcification should be kept in mind when repeat vitreoretinal surgeries are needed in pseudophakic patients.

Visually significant IOL calcification requires IOL explantation. Exchange with a new in-the-bag IOL is usually not possible due to fibrotic capsular bag adhesions, advanced after-cataract and zonular weakening. Therefore, secondary IOLs are usually implanted into the sulcus when the capsule can be preserved or fixed with sutures to the sclera or iris or fixed to the iris using an iris claw lens. IOL exchange with a scleral-fixated type was the preferred technique in most of our patients. Although visual acuity increased significantly and usually reached the best corrected vision before the calcification, it must be kept in mind that the long-term compatibility of such IOLs is not as good as primary in-the-bag implants.

In conclusion, the pathogenesis of secondary IOL calcifications appears to be multifactorial. IOL material, use of intravitreal/gas/air tamponade, combined phacovitrectomy, the presence of air/gas in the anterior chamber, posterior capsulotomy and postoperative inflammation may be identified as risk factors for this complication. Clinically, calcium phosphate deposits developing after vitrectomy share many features with opacifications occurred after anterior segment surgeries, and intraocular inflammation and the presence of gas seem to be of the most significance. Hydrophilic IOLs should be avoided in patients undergoing combined phacovitrectomy as a precaution. IOL explantation and exchange seem to be an effective approach for visual rehabilitation in cases with significant calcifications.

## Figures and Tables

**Figure 1 diagnostics-13-01943-f001:**
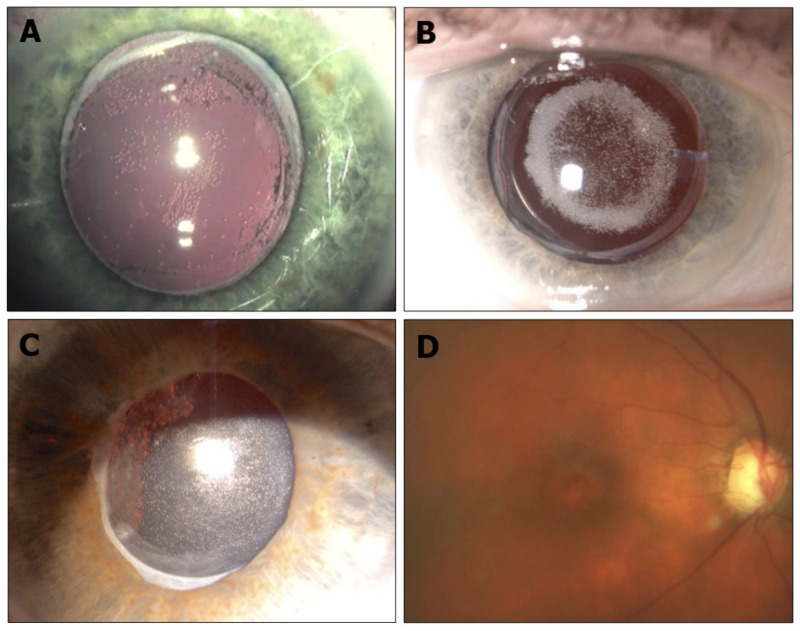
Different densities of superficial opacifications of hydrophilic IOLs in vitrectomized eyes. Treatment included: (**A**) Combined phaco-vitrectomy for proliferative diabetic retinopathy (Acri.Lyc 51LC). (**B**) Vitrectomy and C2F6-gas tamponade combined with phaco/capsular-bag-fixated IOL for retinal detachment (Basis Z). (**C**) Pseudophakic vitrectomy and air tamponade for uveitis (Hydroview H60M). (**D**) fundus image of the same eye (**C**) that shows the clouding effect of the IOL.

**Figure 2 diagnostics-13-01943-f002:**
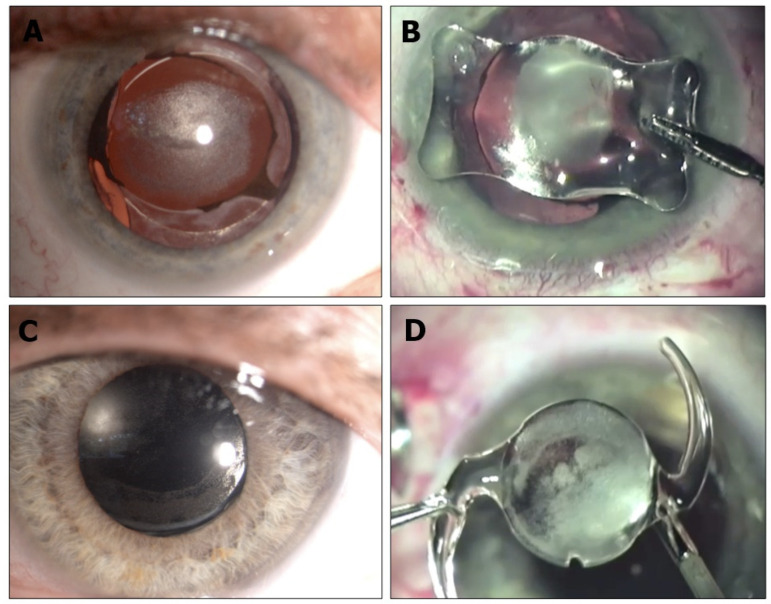
Pre- and intraoperative findings of two eyes that needed IOL exchange for dense IOL calcifications. Phaco-vitrectomy with C2F6-gas was performed for retinal detachment using an AcriSmart IOL (previous Acri.Tec, now Zeiss Asphina) (**A**,**B**—case 4) and Basis Z (1st Q, Mannheim—case 11) (**C**,**D**). Note that the deposits are located in the pupillary area, but the haptics are spared.

**Figure 3 diagnostics-13-01943-f003:**
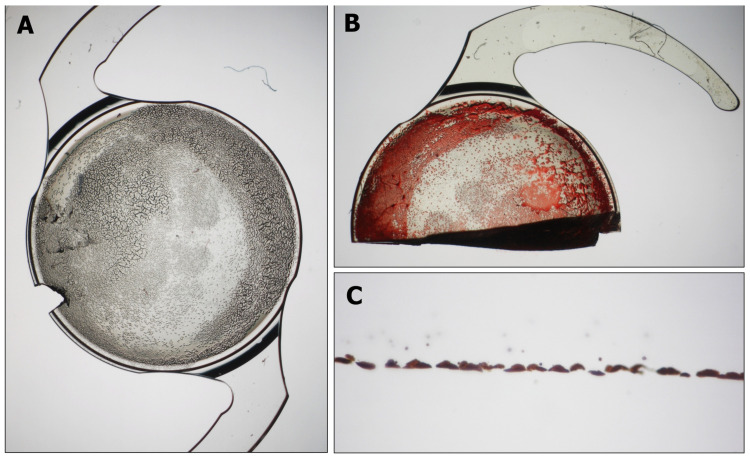
Material analysis of one of the calcified intraocular lenses (case No. 11, B1AW00 Basis Z, 1stQ, Mannheim, Germany): Dense opacification of the anterior IOL optic sparing the haptics (**A**). Alizarin red staining of one half of the IOL stains superficial calcium deposits (**B**). Von Kossa staining of a 5 µm cross section of the IOL reveals calcium deposits underneath the IOL surface within the lens material (**C**).

**Figure 4 diagnostics-13-01943-f004:**
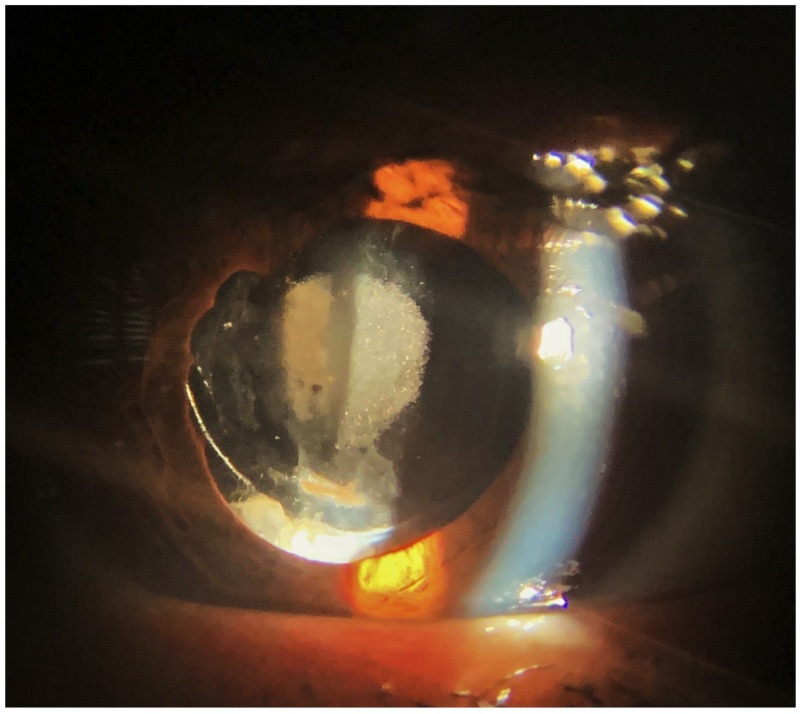
Anterior segment photography of a diabetic patient who underwent combined phaco-vitrectomy with silicone oil tamponade for proliferative diabetic retinopathy. Opacification of the anterior side of the IOL optic was observed, but it did not cause remarkable visual deterioration (BCVA: 20/30). The patient was followed up on without an IOL exchange surgery for two years.

**Table 1 diagnostics-13-01943-t001:** Demographics of the patient cohort.

Patient No.	Age (at IOL İmplantation)	Sex	Systemic Diseases	Date of IOL İmplantation	IOL Type/Manufacturer	IOL Diopters
No. 1	60	Female	HT, CHD	Feb. 2013	CT Asphina 409M (Zeiss)	13.00
No. 2	18	Male	Healthy	May 2016	CT Asphina 409M (Zeiss)	20.00
No. 3	55	Female	HT, HL	Mar. 2014	CT Asphina 409M (Zeiss)	18.50
No. 4	59	Male	HT, CHD	Sep. 2011	CT Asphina 409M (Zeiss)	16.50
No. 5	78	Female	HT, arthritis, allergic rhinitis	Jun. 2012	B1AW00 Basis Z (1stQ)	19.50
No. 6	41	Male	HT	Oct. 2010	C-flex 970C (Rayner)	26.50
No. 7	69	Male	HT, COPD, HL, stroke, renal insuffiency	Jan. 2015	B1AW00 Basis Z (1stQ)	19.50
No. 8	47	Male	Healthy	Apr. 2011	B1AW00 Basis Z (1stQ)	16.50
No. 9	70	Female	HT, arthritis, hypothyroidism	2007	AcriLyc 44S (Zeiss)	19.50
No. 10	71	Female	Cardiac arrhythmia, bronchitis, hypothyroidism	July 2014	Polylens AS61 (Polytech Domilens)	18.00
No. 11	49	Female	Latex allergy	Feb. 2014	B1AW00 Basis Z (1stQ)	19.00
No. 12	42	Male	Healthy	Nov. 2012	Acriva UD613 (VSY)	13.50
No. 13	51	Male	Healthy	Jul. 2009	Acriva UD613 (VSY)	17.00
No. 14	48	Male	Healthy	2008	Unknown	Unknown

HT: Systemic hypertension, CHD: Coronary heart disease, HL: Hyperlipidemia, COPD: Chronic obstructive pulmonary disease.

**Table 2 diagnostics-13-01943-t002:** Clinical and surgical features of patients.

Patient No.	PPV Indication	Date of PPV	Vitrectomy Maneuvers	Post-Op Intra-Camaral Gas	Additional Surgeries (Except IOL Removal or Exchange)	Diagnosis of IOL Opacification
No. 1	Pseudophakic RRD	May 2013	PPV, post. capsulotomy, ILMP with ICG, PFCL, EL, C2F6	Yes	June 2013: Nd:YAG-laser iridotomy for pupillary block	July 2013
No. 2	TRD in uveitis	May 2016	Phaco-IOL, PPV, ILMP, PFCL, EL, silicone oil	No	Aug. 2016: SBS, silicone oil removal, EL, C3F8	Nov. 2016
No. 3	Phakic RRD	Mar. 2014	Phaco-IOL, PPV, ILMP with ICG, PFCL, EL, C2F6	Yes	Mar. 2014: Nd:YAG-laser iridotomy for pupillary block; July 2014: Nd:YAG-laser capsulotomy	May 2016
No. 4	Phakic RRD	Sep. 2011	Phaco-IOL, PPV, ILMP with ICG, PFCL, EL, C2F6	No	None	April 2013
No. 5	Phakic RRD	Jun. 2012	Phaco-IOL, PPV, post. capsulotomy, ILMP with ICG, PFCL, EL, C2F6	No	None	Dec. 2016
No. 6	Pseudophakic RRD	Dec. 2014	PPV, ILMP with ICG, PFCL, EL, C2F6	No	Pre-operative Nd:YAG-laser capsulotomy	Dec. 2015
No. 7	Epiretinal gliosis after PPV	Jan. 2015	Phaco-IOL, PPV, post. capsulotomy, ILMP with ICG, PFCL, EL, air	No	Jan. 2015: Repeat PPV, PFCL, EL, C2F6	Mar. 2017
No. 8	Pseudophakic RRD	Nov. 2013	PPV, post. capsulotomy, ILMP with ICG, PFCL, EL, C2F6	Yes	None	Jan. 2014
No. 9	Phakic RRD	Dec. 2016	Phaco-IOL, PPV, ILMP with ICG, PFCL, EL, C2F6	No data	Repeat PPV with heavy silicone tamponade; silicone oil removal and C2F6 tamponade (in total: four additional surgeries)	No data (surgeries elsewhere)
No. 10	Pseudophakic RRD	Sep. 2016	PPV, EL, heavy silicone oil	Yes	Dec. 2016: Heavy silicone oil—SF6 exchange; gas removal from anterior chamber four days later.	No data (surgeries elsewhere)
No. 11	Phakic RRD	Feb. 2014	Phaco-IOL, PPV, post. capsulotomy, ILMP with ICG, PFCL, EL, C2F6	Yes	Feb. 2014: Anterior chamber revision due to intracameral gas, iris capture and pupillary block	Jan. 2016
No. 12	Pseudophakic RRD	Jan. 2013	PPV, PFCL, EL, air	No	Feb. 2013: repeat PPV, EL, MP, retinectomy, 5000 cs silicone oil; Sep. 2013: silicone–air exchange; Oct. 2013: YAG-laser capsulotomy	Aug 2015
No. 13	Phakic RRD	Jul. 2009	Phaco-IOL, PPV, MP, PFCL, EL, silicone oil	No	Nov. 2009: silicone removal and fluid–air exchange; Nov 2010: Iris fixation of dislocated IOL	May 2011
No. 14	Pseudophakic RRD	May 2015	PPV, PFCL, EL, C3F8	Yes	None	Oct 2015

PPV: pars plana vitrectomy; RRD: rhegmatogenous retinal detachment; post: posterior; ILMP: internal limiting membrane peeling; PFCL: perfluorocarbon liquid; EL: endolaser; Phaco-IOL: phacoemulsification and intraocular lens implantation; ICG: indocyanine green; C2F6: hexafluoroethane; C3F8: perfluoropropane; SF6: Sulfur hexafluoride; SBS: scleral buckling surgery; MP: membrane peeling; IOL: intraocular lens.

**Table 3 diagnostics-13-01943-t003:** Clinical data before and after diagnosis of intraocular lens opacifications.

Patient No.	Date of First Air/Gas Contact to IOL	Opacification First Documented	Explantation Date	Second IOL	BCVA (logMAR) after PPV	BCVA (logMAR) Prior to IOL Explantation	BCVA (logMAR) after IOL Exchange
No. 1	May 2013	July 2013	16 June 2015	Alcon MA50BM (sulcus-fix)	0.20	0.30	0.10
No. 2	Aug. 2016	Nov. 2016	11 December 2016	None (patient left aphakic)	1.30	2.80	2.30
No. 3	Mar. 2014	Aug. 2014	3 May 2016	Avansee PU6AS Kowa (sulcus-fix)	0.40	0.50	0.30
No. 4	Sep. 2011	Apr. 2013	24 January 2017	Avansee PU6AS Kowa (sulcus-fix)	0.10	0.40	0.30
No. 5	June 2012	Dec. 2016	16 January 2017	Avansee PU6AS Kowa (sulcus-fix)	0.30	0.30	0.30
No. 6	Dec. 2014	Dec. 2015	19 July 2016	AMO VRSA54 (Artisan-retropupillary-fix)	0.40	0.40	0.50
No. 7	Jan. 2015	July 2017	9 May 2017	Avansee PU6AS Kowa (sulcus-fix)	0.00	0.10	0.00
No. 8	Nov. 2013	Jan. 2014	8 August 2016	Avansee PU6AS Kowa (sulcus-fix)	0.10	0.50	0.10
No. 9	No data	No data	31 October 2016	Avansee PU6AS (sulcus-fix)	0.70	0.70	1.00
No. 10	No data	No data	18 May 2017	Avansee PU6AS Kowa (sulcus-fix)	n.b.	0.50	0.40
No. 11	Feb. 2014	Jan. 2016	4 March 2016	Alcon MA50BM (sulcus-fix)	0.10	0.30	0.10
No. 12	Jan. 2013	Aug. 2015	5 October 2015	Freedom Lens (sulcus-fix)	1.00	1.30	0.70
No. 13	Nov. 2009	May 2011	15 December 2015	Sensar AR40E (sulcus-fix)	1.00	1.00	0.70
No. 14	May 2015	Oct. 2015	8 November 2015	Alcon MA60AC (sulcus-fix)	0.10	0.40	0.00

IOL: intraocular lens; BCVA: best corrected visual acuity; sulcus-fix: sulcus fixation of intraocular lens; retropupillary fix: retropupillary iris fixation of an artisan-type iris claw lens.

**Table 4 diagnostics-13-01943-t004:** Characteristics of the calcified intraocular lenses.

IOL Brand	Material	Design	No. of Eyes (n: 14)
CT Asphina 409M	Hydrophilic acrylic with hydrophobic coating	Single-piece, plate haptic	4
B1AW00 Basis Z	Hydrophilic acrylic	Single-piece, c-loop haptic	4
Rayner 970C	Hydrophilic acrylic	Single-piece, c loop haptic	1
AcriLyc 44S	Hydrophilic acrylic with hydrophobic coating	Single-piece, plate haptic	1
Polylens AS61	Hydrophilic acrylic	Three-piece, c-loop haptic	1
Acriva UD613	Hydrophilic acrylic with hydrophobic coating	Single-piece, c-loop haptic	2
Unknown	not applicable.	Single-piece, plate haptic	1

## Data Availability

Data presented in the manuscript are available from the corresponding authors upon reasonable request.

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
