# Peer review of "Clinical Characteristics of Patients with Intraocular Lens Calcification after Pars Plana Vitrectomy"

_diagnostics, 2023, doi:10.3390/diagnostics13111943_

Round 1

Reviewer 1 Report

This paper addresses the question of clinical risk factors regarding the opacification of intra-ocular IOLs in patients undergoing pars plana vitrectomy. It is a well written and well illustrated paper showing endo tamponade with hydrophilic IOLs is a particular risk factor.

The conclusions are relevant to the original questions posed.

This is a great paper and very relevant to current practice. 

Author Response

We thank to the reviewer for the nice comments. We checked the manuscript for English language and corrected grammatical mistakes.

Reviewer 2 Report

The authors present an interesting cases series of 14 patients with calcifications of their IOL. The causes are multifactorial and discussed perfectly. This manuscript is of high clinical value and interesting to read. Figures and Tables are informative.

Comments:

Line 211: hydrophilic instead of hydrophobic IOLs

Line 265: what about the diabetic patient in Fig. 4?

Line 272: usually twice, 1 should be enough.

Line 280: appears

It would be interesting to know the (approximately) percentage of ppVEs without any change/calcification of the IOL done in this period 2015-2017. So the readers may estimate the frequency of IOL changes to be expected.

Author Response

We thank to the reviewer for constructive feedback. We corrected our mistakes that the reviewer pointed out.

Reviewer's comment:

Line 216 (formerly 211): hydrophilic instead of hydrophobic IOLs

Author's reply:

We thank to the reviewer for correction. We corrected "hydrophilic" instead of hydrophobic IOLs

Reviewer's comment:

Line 265: what about the diabetic patient in Fig. 4?

Author's reply:

We thank to the reviewer for the valuable question. We could not associate the calcification that developed in the patient with diabetes in Fig 4. In order to make a claim on this subject, it would be very appropriate to plan a study that will look at calcification rates in diabetic and non-diabetic patients.

Reviewer's comment:

Line 272: usually twice, 1 should be enough.

Author's reply:

We thank to the reviewer for correction. We deleted extra "usually".

Reviewer's comment:

Line 280: appears

Author's reply:

We thank to the reviewer for correction. We corrected that error, we also checked and corrected many mistakes like that.

Reviewer's comment:

It would be interesting to know the (approximately) percentage of ppVEs without any change/calcification of the IOL done in this period 2015-2017. So the readers may estimate the frequency of IOL changes to be expected.

Author's reply:

We thank to the reviewer for the constrictive feedback. It would be great to reveal the frequency, but since most of the cases were referred, it was not possible to obtain this data. 

Reviewer 3 Report

The authors reported the clinical characteristics of IOL calcification in 14 vitrectomized eyes and concluded that gas tamponade and hydrophilic IOLs may be risk factors for IOL calcification.

Unfortunately, the results section of this study was inconsistent with the tables. This was unacceptable for a study with such a small sample size, and reduced the trust of readers in the reliability of the data. In addition, this study added little new information.

The following points may be helpful:

1.Table 4 shows that 8 of 14 IOLs had c-loop haptics, whereas line 106 states "7 of 14 IOLs had c-loop haptics and one IOL had prolene haptics". Table 2 shows that 6 eyes had pseudophakic RRD and 6 eyes had phakic RRD, which is inconsistent with line 114 to line 115. The manuscript should be carefully reviewed.

2.Inconsistent decimal points.

3.Missing abbreviations and symbols in the table.

4.No.5 and No.6 had no improvement or even loss of vision after IOL exchange, are there any other complications?

5.Line 155, Please add the percentage of locations with calcium deposits.

6.With small samples and only descriptive statistical analysis, it is not accurate to directly conclude that a certain factor increases the risk of IOL calcification.(Line 183)

Author Response

We sincerely thank the reviewer for constructive criticisms and valuable comments, which were of great help in revising the manuscript. Our responses (AC) to the referee’s comments (RC) are given below.

RC 1. Table 4 shows that 8 of 14 IOLs had c-loop haptics, whereas line 106 states "7 of 14 IOLs had c-loop haptics and one IOL had prolene haptics". Table 2 shows that 6 eyes had pseudophakic RRD and 6 eyes had phakic RRD, which is inconsistent with line 114 to line 115. The manuscript should be carefully reviewed.

AC 1. Thanks for the valuable contribution. Since we wrote the 3-piece IOL separately, we expressed it this way. There are 8 c-loop IOLs in total, but one of them is a 3-piece IOL, so we expressed it this way: "7 of 14 IOLs had c-loop haptics and one IOL had prolene haptics". Many thanks to the reviewer for his valuable correction on the numbers of phakic and pseudophakic RD patients. There was a typo in the article, we changed it to 6 phakic and 6 pseudophakic.

RC 2. Inconsistent decimal points.

AC 2. Thanks for the valuable contribution. We fixed inconsistent decimal points.

RC 3. Missing abbreviations and symbols in the table.

AC 3. Thanks for the valuable contribution. We checked the missing abbreviations and added the missing ones

RC 4. No.5 and No.6 had no improvement or even loss of vision after IOL exchange, are there any other complications?

AC 4. Thanks for the valuable comment. The vision of these patients did not change during the chart evaluation, but improved subjectively.

RC 5. Line 155, Please add the percentage of locations with calcium deposits.

AC 5. Thanks for the valuable contribution. We added the percentage of locations with calcium deposits.

RC 6. Thanks for the valuable comment. With small samples and only descriptive statistical analysis, it is not accurate to directly conclude that a certain factor increases the risk of IOL calcification.(Line 183)

AC 6. The reviewer is very right about this, on line 183 we stated that calcium deposits are the cause of opacification, not the etiological cause.